# Quantification of training load distribution in mixed martial arts athletes: A lack of periodisation and load management

Christopher Kirk[1,2]*, Carl Langan-Evans[1], David R. Clark[1], James P. Morton[1]

**1** Research Institute for Sport and Exercise Sciences, Liverpool John Moores University, Liverpool, United Kingdom, **2** Sport and Human Performance Research Group, Collegiate Crescent, Sheffield Hallam University, Sheffield, United Kingdom

These authors contributed equally to this work.

* C.P.Kirk@2018.LJMU.ac.uk

**Data Availability Statement:** Raw data are available at OSF (DOI: 10.17605/OSF.IO/P9SWU).

**Funding:** The author(s) received no specific funding for this work.

## Abstract

The aim of this study was to quantify typical training load and periodisation practices of MMA athletes. MMA competitors (n = 14; age = 22.4 ± 4.4 years; body mass = 71.3 ± 7.7 kg; stature = 171 ±9.9 cm) were observed during training for 8 consecutive weeks without intervention. Seven athletes were training for competitive bouts whilst the remaining 7 were not. Daily training duration, intensity (RPE), load (sRPE and segRPE), fatigue (short questionnaire of fatigue) and body region soreness (CR10 scale) were recorded. Using Bayesian analyses (BF$_{10}$≥3), data demonstrate that training duration (weekly mean range = 3.9–5.3 hours), sRPE (weekly mean range = 1,287–1,791 AU), strain (weekly mean range = 1,143–1,819 AU), monotony (weekly mean range = 0.63–0.83 AU), fatigue (weekly mean range = 16–20 AU) and soreness did not change within or between weeks. Between weeks monotony (2.3 ± 0.7 AU) supported little variance in weekly training load. There were no differences in any variable between participants who competed and those who did not with the except of the final week before the bout, where an abrupt step taper occurred leading to no between group differences in fatigue. Training intensity distribution corresponding to high, moderate and low was 20, 33 and 47%, respectively. Striking drills accounted for the largest portion of weekly training time (20–32%), with MMA sparring the least (2–7%). Only striking sparring and wrestling sparring displayed statistical weekly differences in duration or load. Athletes reported MMA sparring and wrestling sparring as high intensity (RPE≥7), BJJ sparring, striking sparring and wrestling drills as moderate intensity (RPE 5–6), and striking drills and BJJ drills as low intensity (RPE≤4). We conclude that periodisation of training load was largely absent in this cohort of MMA athletes, as is the case within and between weekly microcycles.

**Competing interests:** The authors have declared that no competing interests exist.

## Introduction

Mixed martial arts (MMA) is a combat sport comprising a unique combination of striking and grappling techniques incorporated from other disciplines including muay Thai, wrestling and Brazilian jiu jitsu (BJJ) [1]. Competitors engage in striking opponents to the head, limbs and torso, using the fists, feet, elbows and knees. Additionally, participants use grappling manoeuvres to attain a more dominant standing or grounded position, with the intention of being able to strike the opponent more effectively, or to cause them to 'submit' to joint locks or chokes. Professional bouts consist of 3 or 5 x 5 minute rounds whilst amateur bouts are 3 x 3 minute rounds [2, 3]. Though data quantifying the direct internal load of competition is currently lacking, the heart rates (HR) of amateur male MMA athletes at the end of each round during simulated bouts was reported as >80% of maximal HR [4]. Such physiological load considered in combination with the requirement to engage in multiple technical performance aspects (striking, wrestling and submission grappling), highlight that the training practices of MMA athletes should incorporate multiple technically and physically demanding components [1, 5]. However, in relation to the latter, it is noteworthy that the completion of MMA technical training in isolation was not associated with improvements in strength, sprint speed, aerobic capacity or limb movement velocity [6]. This apparent lack of a physiological response to solely technical training is especially relevant given that successful and unsuccessful participants may be distinguishable by their lower body force production [7]. The requirement of incorporating a multitude of potentially conflicting technical and physical focused training sessions, coupled with the requirement to "make weight" for competition [8, 9], clearly highlights the challenge and importance of formulating a well-structured and periodised training plan.

In contrast to traditional endurance [10, 11] and team sports [12, 13], a detailed understanding of habitual training practices of MMA athletes has not yet been established. Indeed, reports to date are limited to retrospective questionnaires [14, 15] and a case-study account in which a participant was preparing for a now defunct rule set and competition format [16]. More recent studies do provide the subjective weekly load of MMA pre-competition training [17] and suggest training load ratings for 'hard' and 'easy' days [18]. Whilst these studies measured training load via sessional rating of perceived exertion (sRPE) [19] neither described how these loads are achieved, the intensities of training methods nor how training is modified for a bout. Equally, there is currently no data regarding the fatigue, recovery or resultant soreness of MMA athletes during training periods. As such, current MMA training practice has not been quantified and described in relation to frequency and relative intensities of each training category, periodisation and tapering strategies, nor the associated effects of training on fatigue, recovery and adaptation [20]. Additionally, it is also unclear if training practices are manipulated in those athletes who are actively engaging in body mass reduction strategies in order to 'make weight' for competition [8, 9]. It is therefore difficult to determine which training methods, loading strategies and recovery protocols are most appropriate for this population [21].

Training load can be accurately measured in applied practice via post training RPE which quantifies the gestalt of acute peripheral and central responses to exercise [19, 22]. This method has been applied previously in a range of combat sports to determine training load and response [23, 24]. The sensation of fatigue is due to an array of interdependent, co-affecting systemic changes that occur during exercise and the subsequent recovery period [25–28]. Directly observing this centrally mediated fatigue is therefore invasive and time-consuming. As such, proxy measures of fatigue are recommended in the field [29]. Such methods that have been reported for use in sports include the short questionnaire of fatigue (SQF) [30] and CR10 scales of soreness [31]. Though deemed to be subjective measures, due to the multifactorial

nature of these collection tools, they may well be superior to objective measures in applied settings [32–34].

The void in our understanding of the training practices of MMA may explain the aforementioned absence of physiological adaptations to MMA training [6], as well as the apparent lack of application of research to practice [35]. Therefore, the aim of the study was to quantify the typical training load and periodisation strategies completed by MMA athletes. To this end, we observed a cohort of experienced MMA athletes completing an 8-week training period, during which we quantified training duration, load and associated fatigue and soreness within and between each weekly microcycle, as well as frequency and intensity of each specific training activity completed. Additionally, we also performed a secondary analysis to ascertain if training practices differed between those athletes who were preparing for competition versus those with no upcoming contest. In accordance with traditional periodisation strategies, we hypothesised that training load would be periodised within and between weekly microcycles and that different training patterns would be completed by athletes actively preparing for competition.

## Materials and methods

### Experimental design

Ethical approval was provided by the research ethics committee of Liverpool John Moores University (Ref: 19/SPS/007), with participants providing written, informed consent prior to commencement of data collection. This study utilised a cohort observational study design to record the regular training of competitive MMA participants, without any intervention. The study objective was to characterise any changes in daily and weekly training load and concomitant changes in fatigue and/or body soreness. An additional aim was to determine the relative intensities of MMA training modes and how these are distributed within the week and across the full training period. All MMA training sessions were attended in person by the lead author for the purposes of data collection without intervention to the training sessions themselves. Training data was recorded using a bespoke hand notation system developed through a 6-week period of pilot testing at a single MMA club. Recorded training modes and categories were initially chosen based on the lead author's extensive experience in MMA training and academic awareness of related scientific research. These were then refined further and added to, based on observations of training modes and categories that did not fit those originally defined, and alongside regular discussions with the club coach who confirmed agreement. The final hand notation sheet may be viewed here https://osf.io/p9swu/?view_only= e5d613842a3642cd8324d2141b19de8a. This sheet was utilised to record the content of each session inclusive of categories trained, types of drills trained, time spent on each drill/category (timed using handheld stopwatch) and specific participants taking part in each session. Subjective methods were used due to an absence of validated objective measures related to MMA, and due to evidence supporting these techniques in the field [33].

### Participants

A cohort of 14 competitive MMA participants (participant descriptive data is reported in Table 1) from 4 individual MMA clubs volunteered to take part in this study following written, informed consent and institutional ethical approval based on the following inclusion criteria: aged ≥16 years at the commencement of data collection; taken part in ≥3 official amateur or professional MMA bouts (at least one bout in the 18 months prior to commencement of data collection); be actively training in MMA with the intention of competing in an MMA bout within 6 months of data collection. During data collection 7 participants had competitive

**Table 1. Participant descriptive data.**

| Participant No. | Age (years) | Sex | Stature (cm) | Habitual Body Mass (kg) | Competed? | Pre-bout mass loss (kg and %) | Division Weigh-in Limit (kg) |
|---|---|---|---|---|---|---|---|
| 1 | 21 | M | 178 | 73 | Y | 7.2 (9.9%) | 65.8 |
| 2 | 17 | M | 176 | 76 | Y | 10.2 (13.4%) | 65.8 |
| 3 | 26 | M | 177 | 81 | Y | 10.7 (13.2%) | 70.3 |
| 4 | 18 | M | 186.5 | 72 | Y | 10.8 (15%) | 61.2 |
| 5 | 21 | M | 162.5 | 64 | Y | 7.3 (11.4%) | 56.7 |
| 6 | 19 | M | 188.4 | 84 | Y | 6.9 (8.2%) | 77.1 |
| 7 | 23 | M | 173.5 | 73 | Y | 7.2 (9.9%) | 65.8 |
| 8 | 20 | F | 163 | 66 | N | - | 61.2 |
| 9 | 24 | M | 178 | 75 | N | - | 65.8 |
| 10 | 16 | F | 157 | 72 | N | - | 65.8 |
| 11 | 25 | M | 166.5 | 66 | N | - | 56.7 |
| 12 | 31 | F | 164 | 57 | N | - | 52.2 |
| 13 | 20 | F | 161.5 | 78 | N | - | 70.3 |
| 14 | 29 | F | 163 | 60.5 | N | - | 56.7 |
| **Total Cohort Means** | 22 ± 4.4 | - | 171 ± 9.9 | 71.3 ± 7.7 | - | - | - |
| **Bout means** | 20.7 ± 3.1 | - | 177.4 ± 8.6 | 74.7 ± 6.5 | - | 8.6 ± 1.8 (11.6 ± 2.4%) | - |
| **No-bout means** | 23.6 ± 5.3 | - | 164.7 ± 6.6 | 67.8 ± 7.6 | - | - | - |

Nb. Habitual Body Mass = the participant's body mass prior to commencing pre competition mass loss.

bouts, whilst the other 7 did not. Due to the nature of MMA training sessions, the numbers of people training in addition to the study cohort ranged from 5–40 people per session. This included competitive MMA athletes not involved in the study and non-competitive participants training for fitness and enjoyment. Only the activities of study participants were recorded.

## Procedures

Each of the following procedures were conducted for 9 consecutive weeks: Week 0 was designated for participant familiarisation; Weeks 1–8 was the formal data collection period, during which 405 individual training sessions were recorded and analysed. During Week 0 participants were familiarised with the interpretation and reporting of rating of perceived exertion (RPE), how to complete their short questionnaire of fatigue (SQF) and CR10 soreness rating questionnaire, how to complete their strength and conditioning (S&C) training diary and ask any questions they may have about the procedures. Each Week 0 training session was attended by the lead author and all data was recorded in keeping with the protocols detailed below, to ensure participants were familiar and consistent with the procedures. Particular attention was taken ensuring participants understood the timing requirements and specific anchoring statements of the Foster sessional RPE 0–10 [19], SQF [30] and CR10 soreness [31] scales to ensure data accuracy and validity. No Week 0 data was included in data analysis.

## Training variables

Duration of each session in its entirety, as well as duration spent in each of the training categories defined in Table 2 was recorded to the nearest whole minute, inclusive of rest periods, using a handheld stopwatch. Categorisation of specific drills was confirmed by the club coach at the end of each session to ensure accurate researcher interpretation. Sessions were video recorded for post session review to check data accuracy using a tripod-based camcorder

**Table 2. MMA training category definitions used during data collection.**

| Training Category | Definition |
|---|---|
| Warm up | Any drill or session content specifically aimed at preparing participants to take part in physical activity |
| Striking drills | Any drill consisting of repetition of coach determined striking movements (boxing and/or kickboxing) in groups for the purpose of skill enhancement and/or attainment |
| Wrestling drills | Any drill consisting of repetition of coach determined wrestling movements (taking opponent to the ground or moving yourself from a grounded to a standing position) in groups for the purpose of skill enhancement and/or attainment |
| BJJ drills | Any drill consisting of repetition of coach determined submission grappling movements (either gaining a dominant grounded position or causing the opponent to submit to joint locks and/or chokes) in groups for the purpose of skill enhancement and/or attainment |
| Striking sparring | Live rounds of open skill sparring (boxing and/or kickboxing) designed to put learnt skills into practice in a controlled, non-competitive environment to improve performance. |
| Wrestling sparring | Live rounds of open skill sparring (taking opponent to the ground or moving yourself from a grounded to a standing position) designed to put learnt skills into practice in a controlled, non-competitive environment to improve performance. |
| BJJ sparring | Live rounds of open skill sparring (attempting to submit or attain/hold a dominant position over opponent) designed to put learnt skills into practice in a controlled, non-competitive environment to improve performance. |
| MMA sparring | Live rounds of open skill sparring (full MMA rules) designed to put learnt skills into practice in a controlled, non-competitive environment to improve performance. |
| Circuit training | Any section of a session using repeated MMA skills or muscular endurance exercises for a coach specified time with the intention of improving fitness rather than skill enhancement/attainment. |
| S&C | Any session or section of a technical session used to improve athlete strength, power or endurance. |

Definitions made in agreement with independent MMA coach during pilot testing; Occasions where session sections could fit into more than one category (i.e., striking drills to set up a wrestling takedown) the session coach was asked to state which of the categories they intended the section to be more aimed towards. BJJ = Brazilian jiu-jitsu; MMA = mixed martial arts; S&C = strength and conditioning.

(Panasonic SDR-H81, Osaka, Japan). Where study participants were completing different training sessions simultaneously, or where individual participant session timings differed, these were recorded on the same sheet using annotations to clearly delineate between participant activities and timings. The majority of sessions (84%) took place in the evening between the hours 16:00–21:00, with 16% of sessions taking place between the hours 09:30–13:00. Strength and conditioning (S&C) sessions external to technical sessions were recorded via self-reporting by each participant using a bespoke personal training diary returned to the researchers on a weekly basis.

The perceived intensity of each training category and the session overall was measured by asking each participant individually to record RPE using the Foster sessional RPE 0–10 scale [19] 10–30 minutes after the end of the entire training session [36]. RPE was only collected for categories trained in that session and was measured in arbitrary units (AU). RPE was used to rate intensity of each category using the intensity zone delineations of low (RPE $\leq$ 4), moderate (RPE 5–6) and high (RPE $\geq$ 7) as applied previously [37, 38]. Training load for the full training session was calculated via sessional RPE (sRPE) using the following equation [19]:

$$sRPE \; (AU) = RPE * session \; duration$$

Weekly monotony was also calculated using the following equation [19]:

$$\text{Weekly monotony (AU)} = \text{daily mean sRPE/daily sRPE standard deviation}$$

This equation was modified to calculate between weeks monotony as follows:

$$\text{Between weeks monotony (AU)} = \text{weekly mean sRPE/weekly sRPE standard deviation}$$

Weekly strain was calculated using the following equation [19]:

$$\text{Strain (AU)} = \text{total weekly sRPE} * \text{monotony}$$

Training load for each category was calculated via segmented sessional RPE (segRPE) [39], using the following equation:

$$\text{segRPE (AU)} = \text{RPE} * \text{category duration}$$

## Fatigue and soreness

Immediately after recording RPE data for the final training session of each day, participants completed a paper-based SQF to record their perceived fatigue and wellness [30], with the sum of the responses to each question providing a daily total score for each participant. The weekly mean of each daily score was calculated to provide a weekly fatigue score (AU) for each participant. The timings of training sessions confounded the calculation of test-retest reliability, but SQF reliability has previously been reported as CV = 2.1% [40]. A bespoke, paper-based rating questionnaire recorded participant's perceptions of soreness for the following body regions: head and neck; shoulders and arms; upper torso (upper back and chest); lower torso (lower back and hips); legs. Ratings were based on the following CR10 scale: 0 = no pain; 1–3 = mild discomfort/stiffness; 4–6 = moderate discomfort; 7–9 = noticeable pain; 10 = maximal pain [31]. These body areas were chosen in keeping with the body regions most associated with injuries in MMA competition and training [41]. The soreness questionnaire was completed immediately after the SQF and determined weekly mean soreness for each body region (AU). Completion of both questionnaires took <5 minutes in total. For non-training days, participants were provided a blank set of SQF and soreness questionnaires and were asked to complete these at least 60 minutes before bed. Completed forms were collected from the participants on a weekly basis.

## Statistical analyses

Inference in each of the following tests were based on the calculation of Bayes factors (BF), to provide support for either the hypothesis ($BF_{10}$) or the null hypothesis ($BF_{01}$), respectively. Unless stated, all comparisons were completed using Bayesian repeated measures ANOVA with a default prior r = 0.5, and a default t test with a Cauchy prior as post hoc analysis. Omega squared ($\omega^2$) was calculated as the effect size.

Within week and between week differences in each of the following variables were determined: total session duration, category training duration and sRPE. Category segRPE was compared between weeks. The duration of time spent in each intensity zone was also compared within categories. Between week and between body region differences in soreness was also calculated, as were between week fatigue scores.

Differences in variables between participants who competed and those who did not were also determined. As participants competed at different stages of the 8 week collection period, groups in this test were compared over a 5 week period incorporating 3 weeks prior to the bout (B -3, B -2, B -1), the week of the bout (B 0) and 1 week post bout (B +1).

Drill based categories and sparring based categories were grouped and compared in terms of total time and percentage of time spent in each intensity zone. Between weeks monotony was also compared between those who competed and those who did not. These tests were completed using Bayesian independent samples t tests using a default Jeffery-Zellner-Siow (JZS) prior = .707 [42] and Cohen's d effect size using the standard deviation of the mean scores as the denominator.

The following thresholds were used for each BF: 1–2.9 = anecdotal; 3–9.9 = moderate; 10–29.9 = strong; 30 = 99.9 = very strong; $\geq$ 100 = decisive. Due to default priors being used, BF robustness check were performed. Where a result was found to cross a threshold, both thresholds are reported [42]. For brevity, p values are not reported in the text, but any result found to have BF $\geq$ 3 was also found to have acceptably low probability of type 1 error (p < .05). $\omega^2$ thresholds were set at: very small $\leq$ .01; small $\leq$ .06; medium $\leq$ .14; large > .14. Cohen's d thresholds were set at: small $\geq$ .2; moderate $\geq$ .6; large $\geq$ 1.2; very large $\geq$ 2. Each of the named statistical tests were completed using JASP 0.13.1 (JASP Team, Amsterdam, Netherlands), with data presented as mean ± SD.

## Results

### Quantification of training duration, load, fatigue and soreness between weekly microcycles

There were no differences between the total training duration, sRPE, monotony, strain or fatigue score of any week for the full cohort across all 8 weeks (Fig 1). Between weeks monotony for the full cohort = 2.3 ± 0.7 AU. Additionally, there were no between week differences in body region soreness (Fig 2), but the following post-hoc differences were found between regions: Legs > Head & Neck ($BF_{10}$ = 217); Legs > Upper Torso ($BF_{10}$ = 126,456); Arms & Shoulders > Upper Torso ($BF_{10}$ = 43); Lower Torso > Upper Torso ($BF_{10}$ = 22). There were no differences between regions within weeks.

### Quantification of training duration, load and fatigue between groups

Weekly training duration, sRPE and weekly monotony (Fig 3) were the only variables found to be different between participants who competed and those who did not, but only in the week of the bout (B 0) and the week immediately post bout (B +1). Between week comparisons were only statistically relevant for training duration ($BF_{10}$ = 13, $\omega^2$ = .04; between groups post hoc $BF_{10}$ = 4) and sRPE ($BF_{10}$ = 11, $\omega^2$ = .13; between groups post hoc $BF_{10}$ = 20) when compared between groups. Weekly monotony was also only different between weeks when accounting for group, but with a medium effect and no post hoc differences ($BF_{10}$ = 315, $\omega^2$ = .09; between groups post hoc $BF_{10}$ = 1). There were no statistically relevant differences in strain or total fatigue score either between groups or weeks.

### Quantification of training duration, load, fatigue within each weekly microcycle

When analysed across all seven days, there is a clear trend that weekends have decisively lower training durations than Monday-Friday with medium to large effects (Fig 4). Week 2 is the only timepoint which displays a less than strong statistical difference between all seven days. When excluding weekends, between day differences largely disappear, with only weeks 1, 3 and 6 displaying greater than anecdotal differences with very small to medium effects. Statistically relevant post-hoc differences were almost entirely between midweek days and weekend days ($BF_{10}$ = 3–1,225). Midweek post-hoc differences in training duration were limited to

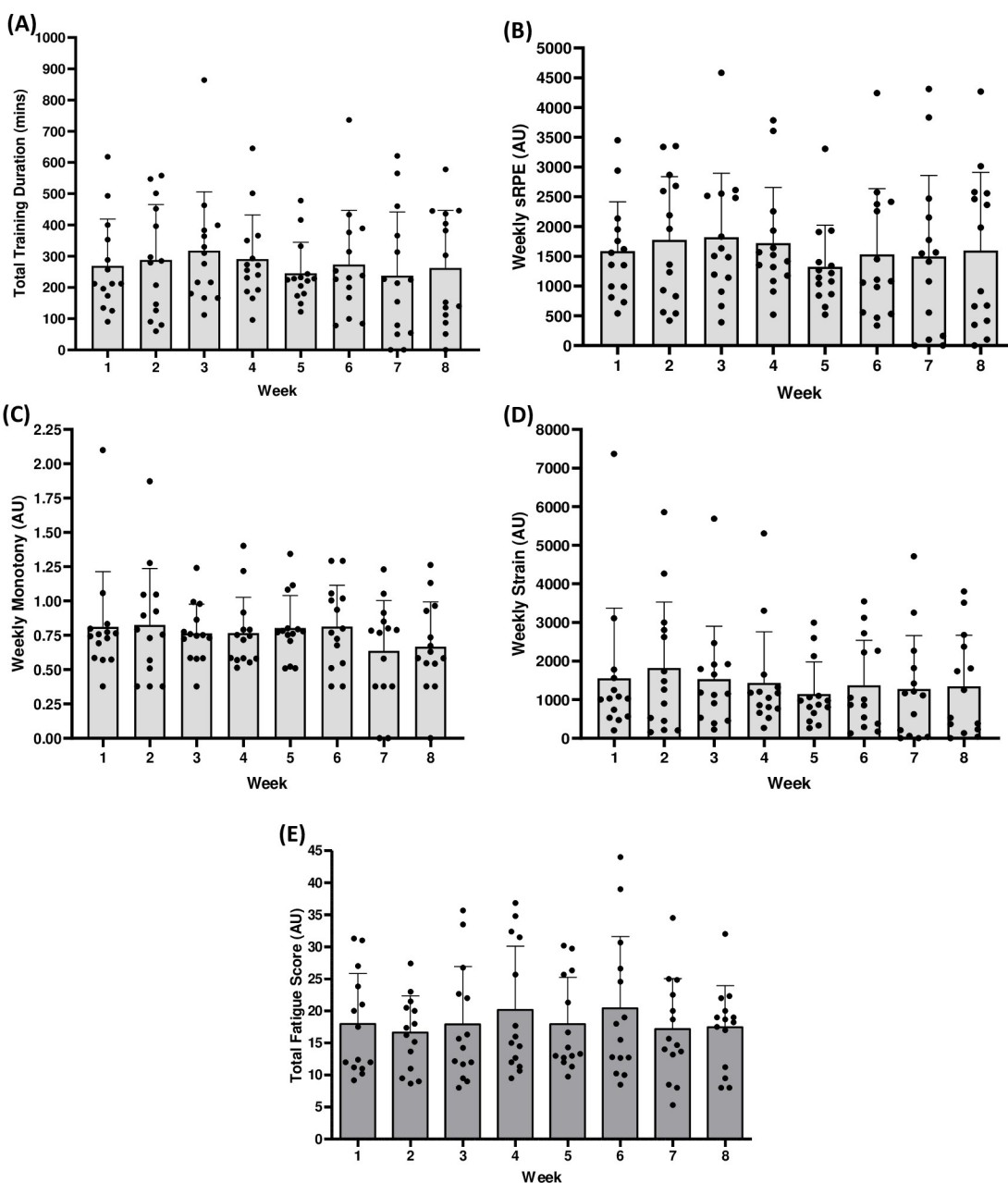

**Fig 1. No between week differences in training duration (mins).** (a), sRPE (b), monotony (c), strain (d) or total fatigue score (e) (all AU unless stated). Black dots represent individual participants. Error bars = SD.

week 1 (Monday > Tuesday $BF_{10}$ = 17; Tuesday < Wednesday $BF_{10}$ = 4) and week 3 (Friday < Monday $BF_{10}$ = 407; Friday < Wednesday $BF_{10}$ = 26). Similarly, there was a consistent statistical difference between the sRPE of midweek days and weekend days with large effects (Fig 5). This difference was, however, absent in week 2. Post-hoc differences between sRPE of midweek days and weekend days had range $BF_{10}$ = 3–563. Monday-Friday between day differences were only found in week 1 (Monday > Tuesday $BF_{10}$ = 11) and week 3 (Monday > Friday $BF_{10}$ = 34; Wednesday > Friday $BF_{10}$ = 15) respectively.

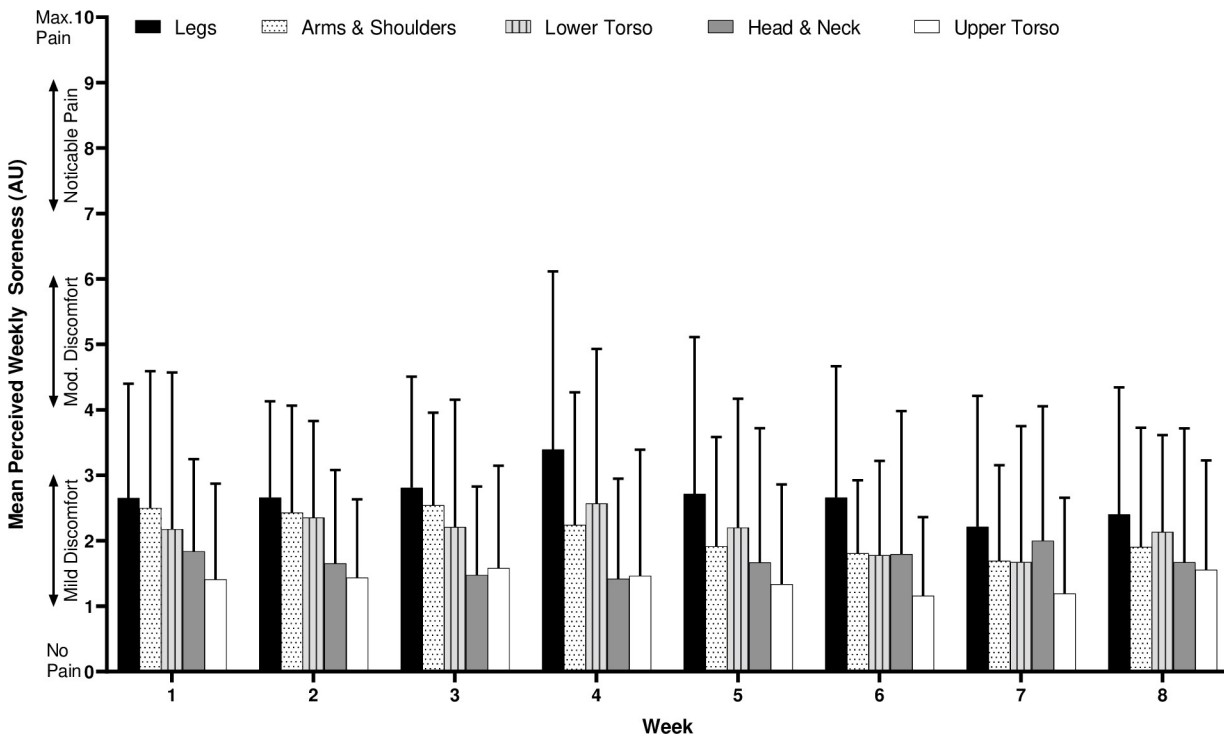

**Fig 2. Between and within week comparisons of perceived body region soreness (AU).** See accompanying text for statistical comparisons between regions. Error bars = SD.

## Training category duration

Table 3 displays descriptive data for duration of each training category by week, and the mean duration of each category per session. Between category durations were found to be decisively different with a large effect ($BF_{10}$ = 4.254e$^{+113}$, $\omega^2$ = .36). Warm-up duration was shorter than all other categories with the exception of wrestling sparring and circuit training ($BF_{10}$ = 6,746–1.807e$^{+86}$). More time per session was spent on striking drills ($BF_{10}$ = 2,970) and BJJ drills ($BF_{10}$ = 216,437) than wrestling drills, with no difference between striking drills and BJJ drills. Within sparring modes, BJJ consisted of longer durations than striking ($BF_{10}$ = 1,683) and wrestling ($BF_{10}$ = 156,504). More time was spent on MMA sparring than striking ($BF_{10}$ = 106,151) and wrestling sparring ($BF_{10}$ = 2.423e$^{+6}$) and with no difference in comparison to BJJ sparring. More minutes per session tended to be spent on technical drills than sparring ($BF_{10}$ = 7–6.599e$^{+17}$), with the exception of wrestling drills, which had no statistical differences to BJJ sparring or MMA sparring, respectively. Where participants took part in S&C, they were of longer duration than all other categories ($BF_{10}$ = 3.978e$^{+8}$–1.807e$^{+86}$). Only wrestling sparring displayed different durations between weeks ($BF_{10}$ = 12, $\omega^2$ = .13), with week 3 being lower than weeks 2, 5 and 8 ($BF_{10}$ = 8–24). Similarly, week 7 had a lower wrestling sparring duration than weeks 2 and 8 ($BF_{10}$ = 3–5).

## Segmented sessional RPE of training categories

Table 3 also displays the weekly mean segRPE for each training category. Decisive differences were found between the segRPE of training categories ($BF_{10}$ = 3.725e$^{+128}$, $\omega^2$ = .39). Post hoc analyses found warm up to cause lower segRPE than all other categories ($BF_{10}$ = 6.980e$^{+9}$–1.335e$^{+85}$). In terms of technical categories, striking drills produced greater load than wrestling

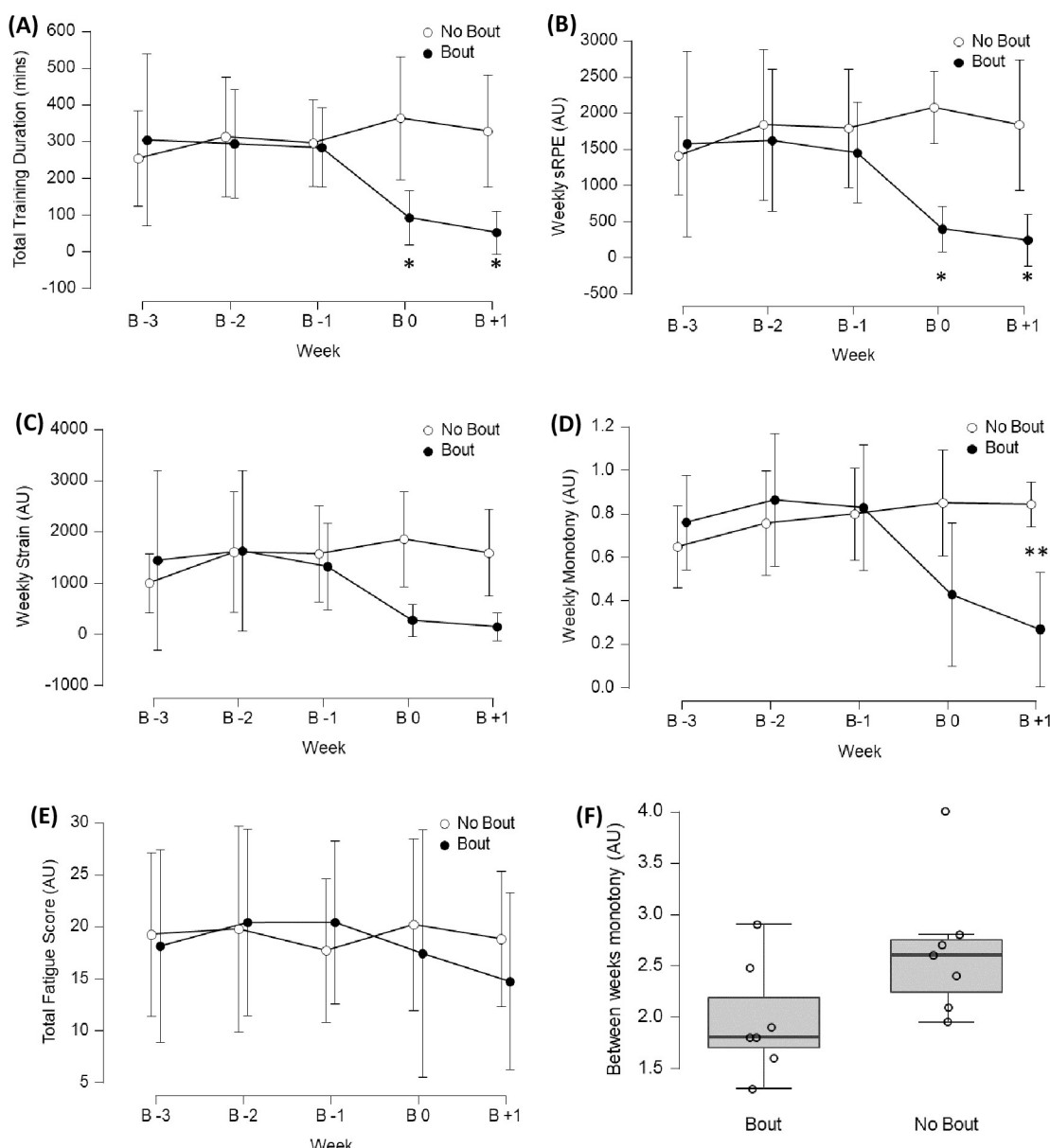

**Fig 3.** Plots A-E: Between week comparisons of training load variables and fatigue score split by bout and no bout. B = weeks before/after bout; * between weeks and between groups differences; ** between weeks differences only, Error bars = 95% credible intervals. Plot 3F: comparison of between weeks monotony between groups, Median [10–90%].

drills ($BF_{10}$ = 14) with no other differences between categories. BJJ sparring produced greater load than striking ($BF_{10}$ = 651) and wrestling sparring ($BF_{10}$ = 992), without any relevant difference to MMA sparring. Wrestling sparring also caused lower load than MMA sparring ($BF_{10}$ = 1.240e$^{+6}$). There was a general trend for technical drills to cause a greater load than sparring, with striking drills segRPE being decisively greater than striking ($BF_{10}$ = 11,167) and wrestling sparring ($BF_{10}$ = 25,578). BJJ drills also induced greater loads than striking sparring ($BF_{10}$ = 21,540) and wrestling sparring ($BF_{10}$ = 24,453). Only MMA sparring was found to elicit statistically greater loads than wrestling drills ($BF_{10}$ = 217). S&C segRPE was found to be greater than all other categories ($BF_{10}$ = 14–1.335e$^{+85}$). Wrestling sparring load was found to

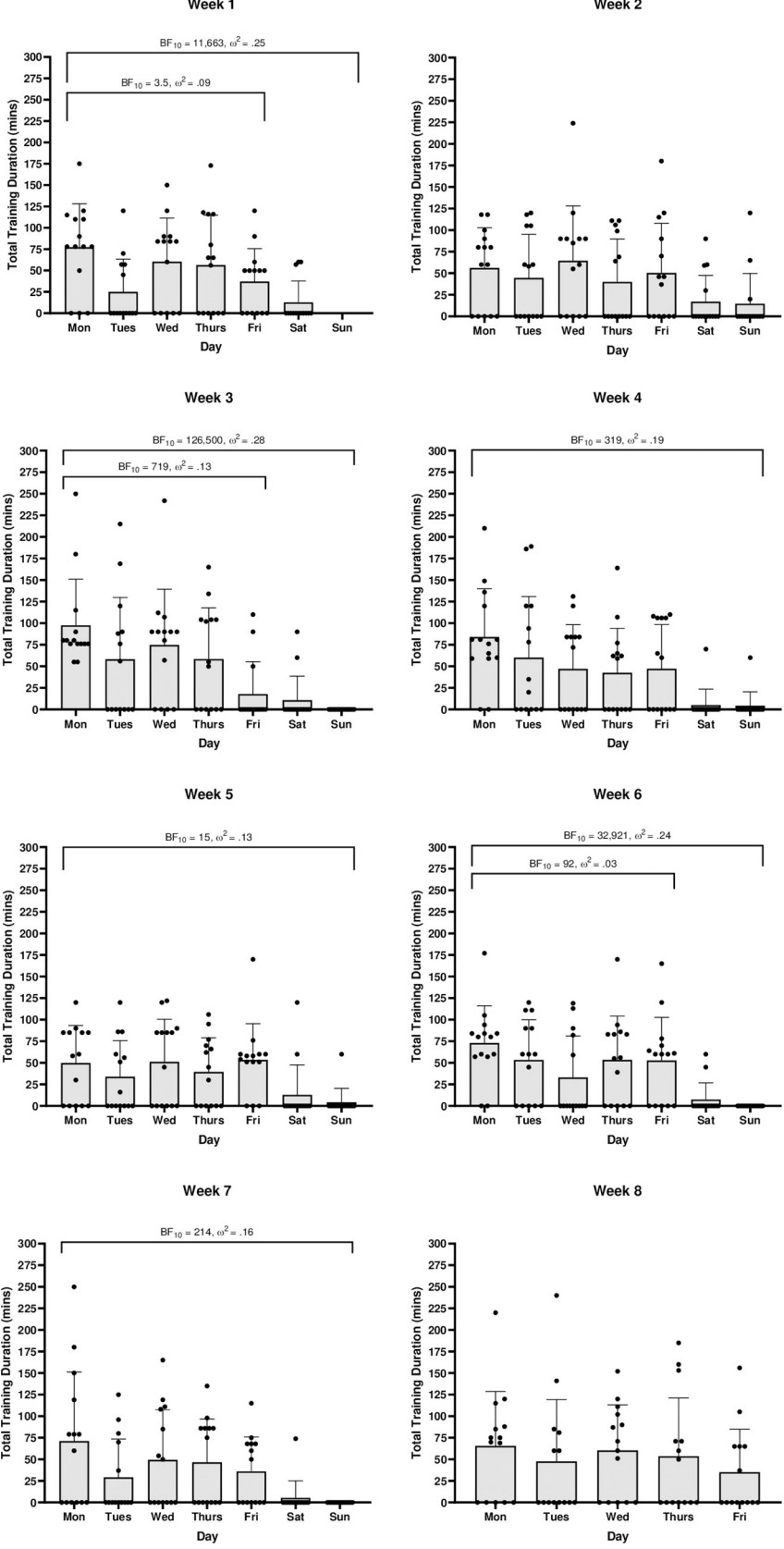

**Fig 4. Within week training duration (mins) comparisons.** Black dots represent individual participants. Error bars = SD.

differ between weeks ($BF_{10}$ = 5, $\omega^2$ = .11), due to week 3 segRPE being lower than weeks 2, 5 and 8 ($BF_{10}$ = 4–34). Striking sparring segRPE also differed between weeks ($BF_{10}$ = 3, $\omega^2$ = .11). In this category week 3 had lower load than weeks 1, 2 and 8 ($BF_{10}$ = 6–504). Week 8 also had lower striking sparring segRPE than weeks 1, 2, 5 and 7 ($BF_{10}$ = 6–1,430). Finally, week 6 was lower than week 5 ($BF_{10}$ = 3) whilst week 7 was lower than week 6 ($BF_{10}$ = 6).

## Training category intensities

When analysing RPE as a marker of training category intensity (Fig 6A), there were decisive differences with a large effect between categories ($BF_{10}$ = 1.168e$^{+134}$, $\omega^2$ = .40). Warm-ups were of lesser intensity than all other categories ($BF_{10}$ = 1.882e$^{+9}$–2.245e$^{+64}$). Within drill-based categories wrestling was found to be more intense than striking ($BF_{10}$ = 5) and BJJ ($BF_{10}$ = 59). Striking sparring was perceived to be of lower intensity than wrestling ($BF_{10}$ = 137) and MMA sparring ($BF_{10}$ = 986). BJJ sparring was also less intense than MMA sparring ($BF_{10}$ = 10) with no other between sparring category differences. All technical drills were recorded as lower intensity than all sparring categories ($BF_{10}$ = 2,436–7.723e$^{+21}$). Circuit training caused a greater RPE than all categories with the exception of BJJ, wrestling and MMA sparring ($BF_{10}$ = 12–7.918e$^{+26}$), whilst S&C was more intense than all categories with the exception of wrestling sparring, MMA sparring and circuit training ($BF_{10}$ = 15–3.271e$^{+49}$).

## Training intensity zones

MMA related total training durations were categorised as 47% at low intensity, 33% at moderate intensity and 20% at high intensity. Fig 6B displays the mean duration of time spent in each intensity zone for each MMA training category. Drill based categories consisted of greater amounts of total time at low intensity in comparison to sparring based categories across the full 8 weeks with a very large effect ($BF_{10}$ = 8–12, d = 3.1). Though there was a large effect between drill based and sparring based categories in terms of mean time per session at low intensity, the evidence for this was only moderate ($BF_{10}$ = 3, d = 2.2). Post-hoc ANOVA differences were found between durations spent at low and high intensities in warm up ($BF_{10}$ = 155,619), BJJ drills ($BF_{10}$ = 228), wrestling sparring ($BF_{10}$ = 3) and MMA sparring ($BF_{10}$ = 10). BJJ drills also displayed post-hoc differences between moderate and high intensities ($BF_{10}$ = 9), with striking sparring having differences between low and moderate ($BF_{10}$ = 37). Fig 6C shows the percentage of time spent in each intensity zone for each MMA category. The percentage of total time spent in low intensity was greater in drill categories than sparring categories ($BF_{10}$ = 23–37, d = 4.1). The only differences between drills and sparring at high intensity was in the percentage of total time spent in this zone ($BF_{10}$ = 11, d = 3.3).

## Discussion

The aim of this study was to apply accepted methods of training load and fatigue monitoring in MMA to describe the training practices of this growing population of athletes for the first time. We report novel data that quantifies typical training load and periodisation strategies employed by MMA athletes. In contrast to our hypothesis, we observed limited evidence of training periodisation within or between weekly microcycles. Additionally, differences in training load practices between athletes preparing for competition and those in normal training was limited to the final week before competition, largely reflective of reduced training

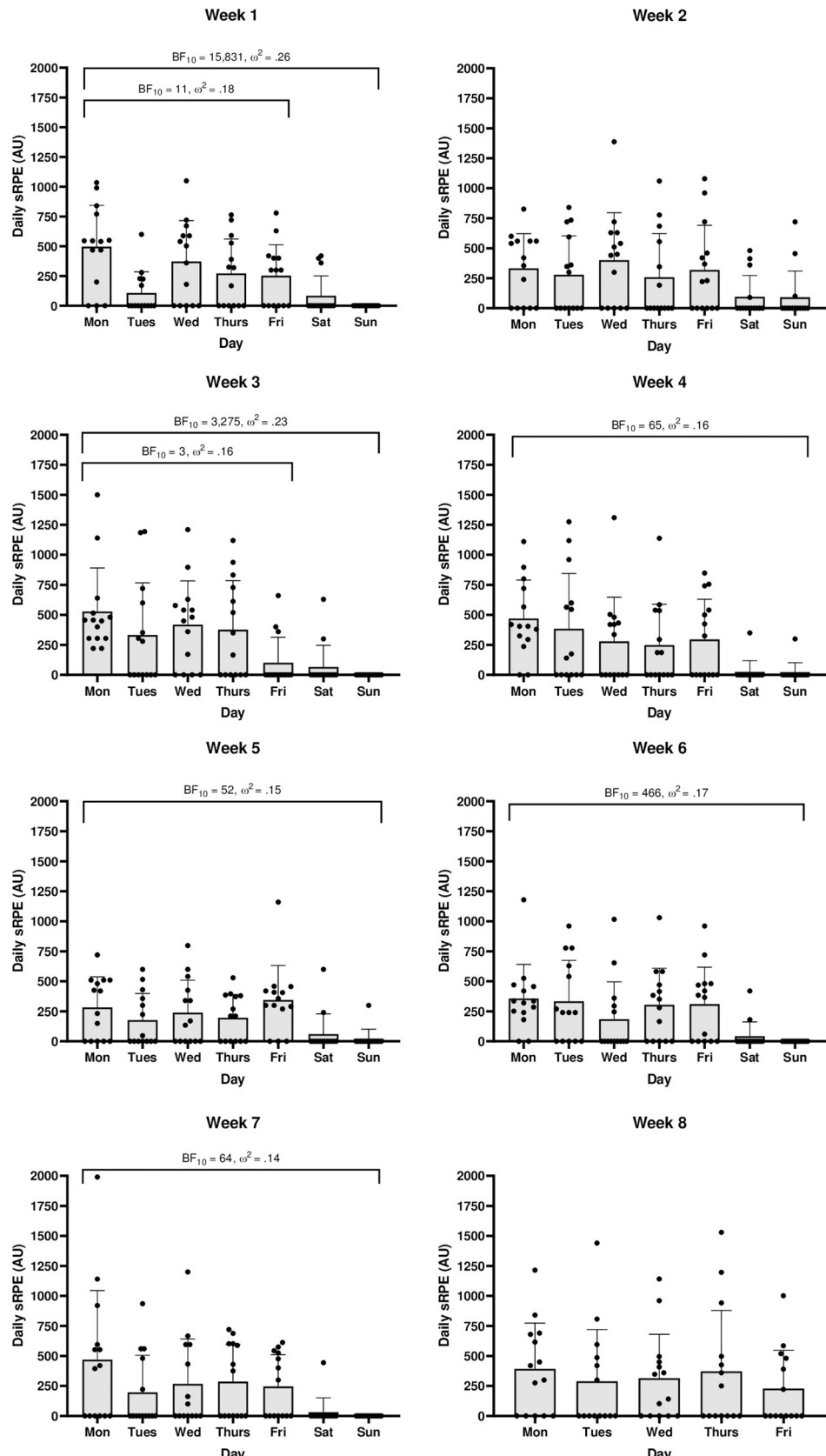

**Fig 5. Within week sRPE (AU) comparisons.** Black dots represent individual participants. Error bars = SD.

**Table 3. Category duration (mins) and segmented sessional RPE (AU) per week and mean per session.**

| | | 1 | 2 | 3 | 4 | 5 | 6 | 7 | 8 | Category Means |
|---|---|---|---|---|---|---|---|---|---|---|
| Warm Up | Duration | 10.2 ± 4.2 (11%) | 11.2 ± 4.6 (11.4%) | 9.5 ± 4 (9%) | 10.2 ± 4.5 (10.7%) | 10.4 ± 3.6 (10.9%) | 10 ± 7.3 (11.1%) | 7.8 ± 3.7 (8.8%) | 10.9 ± 4 (11.8%) | 10 ± 4.7 |
| | segRPE | 28.9 ± 20.2 | 36.7 ± 23.2 | 31 ± 22.3 | 28.1 ± 21.4 | 30.4 ± 16.2 | 32.8 ± 31.9 | 26.7 ± 18.1 | 39.7 ± 19.7 | 31.8 ± 22.3 |
| Wrestling Drills | Duration | 18.3 ± 11.9 (13.1%) | 17.1 ± 11.8 (6%) | 21 ± 11.9 (11.3%) | 29.7 ± 14.8 (15.1%) | 19.5 ± 12.4 (9.9%) | 17.5 ± 16.5 (8.1%) | 18.9 ± 10.8 (11%) | 19.5 ± 11.6 (10.6%) | 20.3 ± 13 |
| | segRPE | 86.4 ± 59.6 | 76.8 ± 59.4 | 116.9 ± 86.8 | 136.1 ± 69.2 | 78.7 ± 47.8 | 80.9 ± 91.6 | 87.3 ± 50.6 | 105.5 ± 73 | 97.4 ± 70.3 |
| Striking Drills | Duration | 33 ± 18.9 (29.6%) | 30.1 ± 18.3 (27.6%) | 32.3 ± 21.5 (29%) | 21.3 ± 16.1 (19.8%) | 30.7 ± 19.8 (29.9%) | 29.9 ± 20.8 (32.4%) | 27.3 ± 23.5 (24.3%) | 26.8 ± 20.9 (23.3%) | 29.1 ± 20 |
| | segRPE | 152.4 ± 99.2 | 149.7 ± 111.3 | 125.5 ± 99 | 82.7 ± 70.9 | 135.5 ± 101.6 | 127.1 ± 100.1 | 128.9 ± 137.1 | 130.6 ± 107.3 | 128.8 ± 104.2 |
| BJJ Drills | Duration | 24.7 ± 15 (10.3%) | 27.7 ± 14.7 (12.1%) | 31.2 ± 14.8 (16%) | 27.4 ± 11.9 (11.5%) | 38 ± 9.4 (13.5%) | 39.6 ± 14.7 (13.5%) | 30.6 ± 15.9 (16.7%) | 27.6 ± 18.3 (17.7%) | 30.1 ± 15.2 |
| | segRPE | 106 ± 97.5 | 107.3 ± 70.7 | 122.8 ± 55.7 | 124.7 ± 65 | 121.1 ± 55.9 | 136.4 ± 52.5 | 120.6 ± 80.6 | 109.9 ± 73.5 | 117.8 ± 69.5 |
| Striking Sparring | Duration | 14.5 ± 2.6 (5.2%) | 14.3 ± 3.7 (4.2%) | 11.3 ± 1.9 (2.9%) | 14.7 ± 11.3 (9.3%) | 15.9 ± 9.4 (6.4%) | 9.7 ± 4.2 (4.2%) | 14.8 ± 5.6 (6.1%) | 8.4 ± 3.3 (3.3%) | 13.1 ± 7.1 |
| | segRPE [b] | 83 ± 15.6 | 90.3 ± 39.3 | 51.8 ± 11 | 85.4 ± 73.2 | 103.6 ± 69.6 | 53.7 ± 25.8 | 88.8 ± 35.4 | 43.42 ± 18.41 | 75.7 ± 49.1 |
| BJJ Sparring | Duration | 23.7 ± 7.7 (7.8%) | 21 ± 19 (12.2%) | 20.6 ± 14.1 (14.2%) | 22.8 ± 13.6 (10.9%) | 16.5 ± 9.8 (8.4%) | 14.5 ± 6.8 (7.7%) | 19.3 ± 9.9 (9.9%) | 15.8 ± 8.1 (8.2%) | 19.3 ± 12.4 |
| | segRPE | 157 ± 79.3 | 128.1 ± 121.8 | 126.6 ± 85 | 137.9 ± 124.5 | 89.3 ± 69 | 88.8 ± 54.5 | 118.1 ± 74.7 | 98.1 ± 70.2 | 117.5 ± 89.9 |
| Wrestling Sparring | Duration [a] | 11.1 ± 5.4 (4.6%) | 17.5 ± 11.5 (5.6%) | 8.1 ± 5 (5.4%) | 10.2 ± 6.3 (6.1%) | 19.7 ± 17.5 (6%) | 10.2 ± 7.9 (2.9%) | 7.6 ± 2.5 (2.3%) | 13.8 ± 6.2 (4.9%) | 11.5 ± 8.6 |
| | segRPE [b] | 70.5 ± 37.9 | 118.9 ± 78.4 | 50.8 ± 36.3 | 67.6 ± 40.8 | 103 ± 81.3 | 68.9 ± 66.3 | 55.7 ± 23 | 99.6 ± 66.4 | 74.4 ± 55.8 |
| MMA Sparring | Duration | 27.7 ± 18 (7.4%) | 18.7 ± 13.5 (5.5%) | 18.8 ± 2.1 (1.9%) | 21.3 ± 5.1 (3.8%) | 21.1 ± 6.1 (5.7%) | 16.6 ± 13.2 (5.6%) | 29.6 ± 13.1 (7.1%) | 19.3 ± 11 (5%) | 21.4 ± 12.4 |
| | segRPE | 181.3 ± 135.3 | 142 ± 126.5 | 111.8 ± 12.8 | 150.5 ± 42.9 | 132.4 ± 40.3 | 102.3 ± 80.6 | 210.3 ± 127.7 | 143.5 ± 99.5 | 146.1 ± 99.3 |
| Circuit Training | Duration | 10.3 ± 6.9 (1.2%) | 10.1 ± 6.1 (2.1%) | 11.3 ± 11 (0.9%) | 5* (0.2%) | 22.3 ± 19.7 (2.3%) | 9 ± 1.4 (0.6%) | 0 (0%) | 9.8 ± 9.2 (1.3%) | 11.5 ± 9.5 |
| | segRPE | 70.3 ± 49.3 | 65.4 ± 37.3 | 58.3 ± 53.5 | 50* | 157 ± 138 | 47 ± 32.5 | 0 | 81.5 ± 82.8 | 77.3 ± 67.6 |
| Strength & Conditioning | Duration | 53.3 ± 7.5 (9.5%) | 50.6 ± 15.5 (13.3%) | 61.7 ± 16 (9.5%) | 42.2 ± 31.8 (12.6%) | 45.2 ± 18.7 (9.1%) | 50 ± 19.9 (13.9%) | 45.2 ± 19.8 (13.9%) | 55.6 ± 29.1 (14.3%) | 49.8 ± 21.5 |
| | segRPE | 360 ± 121.9 | 343. ± 158.1 | 411.7 ± 160.1 | 301.9 ± 247.9 | 333 ± 193.6 | 311.6 ± 187.2 | 343.1 ± 200.5 | 383.8 ± 184.2 | 344.4 ± 181.9 |

Duration data shown as within week group mean ± SD minutes and (% of total weekly training time spent in category);

* only one occurrence of this category in this week;

a = strong $BF_{10}$ and medium $\omega^2$ between weeks;

b = moderate $BF_{10}$ and medium $\omega^2$ between weeks; RPE = rating of perceived exertion (AU).

duration. Related to this limited evidence of periodisation, we observed no changes in ratings of fatigue or soreness throughout the 8-week observational period. From a practical perspective, we also provide novel data by reporting the perceived intensities associated with the full spectra of training activities habitually completed by MMA athletes. As such, this data may provide a platform to develop subsequent sport specific training periodisation strategies whilst potentially explaining the absence of physiological adaptations to MMA training previously reported [6].

Despite the well evidenced role of training load periodisation [43], the training observed in the present cohort of athletes did not change between weekly microcycles. Reported here for the first time, the weekly training duration of ~3–6 hours per week is less than that reported in judo [44] and boxing [45], and only half the weekly duration reported in non-combat sports [10, 13]. Each of these sports use changes in training duration to manipulate overall load, a method that appears to be unused in this cohort resulting in no changes in weekly load. It may

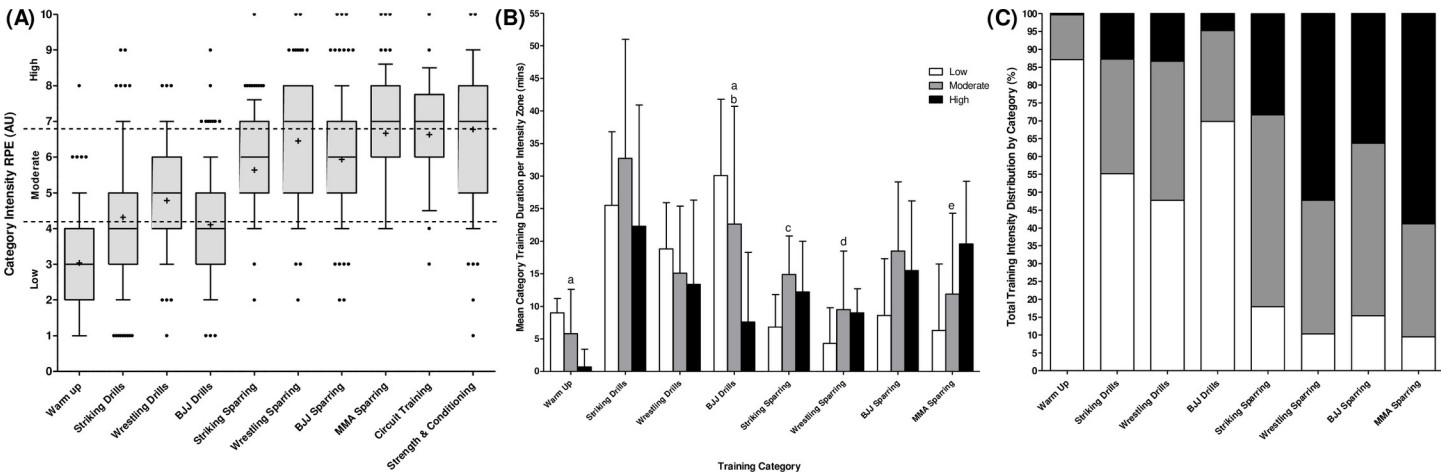

**Fig 6.** A) Median [10–90%] rating of perceived exertion per category (AU); B) Mean time spent in each intensity zone per MMA training category (mins), Error bars = SD; C) Percentage of total time spent in each intensity zone per MMA training category. 6a. Dots show outliers in each category; + = category mean. 6b. a = decisive post-hoc differences between low and high intensity; b = moderate post-hoc differences between moderate and high intensity; c = very strong post-hoc differences between low and moderate intensity; d = moderate post-hoc differences between low and high intensity; e = strong post-hoc differences between low and high intensity.

be that MMA coaches are constrained in planning session duration around the expectations of their paying customers rather than the needs of their competitive athletes, who were in the minority in the observed sessions. In this light, MMA coaches may require more specialised methods of load manipulation to account for this relatively unique circumstance. Methods developed should allow competitive participants to experience overloading but still allow recreational participants to train alongside them. The absence of periodisation is also apparent during competition preparation, where effective tapering strategies were absent. Over the 5 comparable weeks, the only reduction in training duration or load was seen in the week of the bout. Rather than employing an exponential taper over the final 14 days [43] participants reduced training load by more than 2/3 in an abrupt stepwise manner 7 days prior. With no comparable reduction in fatigue in the week of the bout or the week after, it is possible that this approach did not improve athlete readiness or performance. This result is supported by recent data from another research group who also found that MMA training load is only reduced in the week of the bout [17]. In addition to extreme 'weight cutting' seen in our cohort and MMA competitors in general [8, 9], it is more likely that performance would have been impaired. Though we did not collect any direct performance data, evidence from boxing suggests that a taper of 10 days or fewer causes a reduction in combat sport performance until several days after competition [45].

Planned within week undulation of load and duration is a common practice in individual sports without regular competition to achieve overall weekly training fluctuations [45–47]. The absence of this practice in our data results in the static load between weeks and may also explain the lack of changes in fatigue. Comparisons to other sports show daily sRPE of MMA is low [48, 49] and most days would be classed as 'easy' according to the arbitrary definition recently suggested for MMA [18]. The daily load provided therefore may not be great enough to cause sufficient strain to bring about fatigue beyond the acute stage, reducing the likelihood of optimal fatigue-recovery-adaptation. Equally, the minimal training at weekends may be deliberate to allow post-training soreness and fatigue to dissipate enough to remain mild-moderate in the long term [21, 29]. Whilst this reduces the chances of non-functional overreaching (NFOR) it is also unlikely to be sufficient to cause the desired physiological adaptations [20] and therefore improve performance. Effective periodization of training duration and load

structure specific to the needs and environment of MMA may allow coaches to plan training with sufficient daily variation to maintain health and improve performance.

The lack of changes in body region soreness may have a number of explanations. Due to repeated physical impact of MMA, participants may be conditioned against the effects of such strain on musculotendon soreness [50, 51]. This repeated exposure may also have caused a 'normalising' of perceived soreness as 'part of the sport' [52], resulting in higher individual thresholds for 'moderate discomfort' and 'noticeable pain'. Though this cohort displayed similar lower body soreness to rugby players [53], this occurred without the elevated soreness to the upper torso or head and neck regions that is common to impact and grappling inclusive sports [53]. Due to anecdotal experience and extant injury data [41], this result was unexpected and may indicate purposeful planning on the part of coaches to minimise soreness by keeping overall training load low [48]. As the effects of MMA on musculotendon structures are currently unknown, determining the effects of different MMA training loads, durations and practices on this aspect of athlete preparation may be an important area of research moving forward.

Given MMA is a complex sport with multiple conflicting training demands, the need for coaches to ensure their athletes are sufficiently trained in each technical area is reflected in our training category data. BJJ drills had the longest average duration, with the shortest being wrestling sparring. MMA sparring caused the greatest average training load overall, and the highest or second highest training load in 6 of the 8 weeks. Striking drills was the second most load inducing category overall and was the greatest or second greatest load inducing category in 5 of the 8 weeks with the second longest average duration. Despite these weekly variations in category durations, only one was statistically relevant. This absence of weekly changes in category use likely contributes to the static weekly and daily loads reported. Coaches do, however, appear to attempt to manage intensity and load within sessions. Though wrestling sparring and MMA sparring were perceived as the two most intense categories, the former had the second shortest average duration, with the latter being of relatively moderate duration. Conversely, the categories displaying the longest average durations—striking drills and BJJ drills— were also the two least intense. This may indicate coaches are cognizant of too much high intensity work in one session and the negative associated consequences [54, 55]. It does appear though that this is led by pre-conceived notions of category intensities. More time being spent on BJJ sparring than striking sparring, for example, may be evidence that coaches perceive BJJ to be lower intensity. According to our data, these two categories are actually of equal intensity and spending more time on BJJ sparring leads to a greater overall load from this category. Additionally, though striking drills are generally low perceived intensity, the high amount of time spent training this category leads to the highest average time spent at moderate intensity, and an equal amount of high intensity time as MMA sparring. It therefore appears that planning training duration based on anecdotal conceptions of category intensity may not be adequate for balancing training demands. Equally, attempting to train each category to the same extent each week may be the cause of the static weekly training load and fatigue observed.

The current cohort of MMA athletes perceived 80% of their training time as low-to-moderate intensities with 20% as high-intensity. This may be an appropriate split for these athletes considering the physiological responses to these intensity zones [49]. However, little of the training conducted at low intensity was performed at steady-state, owing to these sessions consisting of skill orientated, intermittent drills with prolonged rest periods, likely reducing the aerobic adaptations from these sessions. Compounding this is the low number of S&C sessions reported by this cohort (63 out of 405 individual sessions), none of which could be qualitatively described as steady-state or aerobic endurance. Given MMA training alone is insufficient at improving aerobic or anaerobic performance [6], in-bout reductions in pacing are to be

expected [56]. Though the intensity distribution observed here may potentially be appropriate for the unique physiological requirements of combat sports, the absence of supplementary conditioning across the cohort in general likely negates its effectiveness. Based on the presented data, therefore, it may be possible to plan session content and intensity on a weekly basis. In this hypothetical structure, it may be the case that low load weeks consist entirely of striking and BJJ drills, with only one day of wrestling drills or wrestling sparring. High load weeks might consist of one day of striking or BJJ drills, with several days of wrestling, BJJ and MMA sparring. This would allow clear delineation between weeks of high, moderate and low load, enabling functional overreaching to occur followed by restitution weeks [43]. This process may elicit the physiological improvements associated with spending <10% of weekly and total training time at high intensity [57], alongside the performance benefits of shock or overloading weeks [58]. This may allow different weeks to focus more time on specific categories to equalise the balance between skill transfer and physiological adaptation [59], whilst still satisfying the aforementioned duration expectations of paying customers. The category intensity ratings provided here may allow coaches to start planning training in this manner and researchers to more accurately quantify training responses.

Our data do have some potential limitations. The cohort studied were mostly high-level amateurs. The training practices of high-level professionals may be different, however, restrictions on duration and content caused by MMA training sessions including recreational paying customers is a common feature across the sport for professionals and amateurs alike, outside of a very small number of large training centres [60]. It is also common in MMA for even professional competitors to work in paid employment alongside their training, restricting the amount and frequency of training in a similar manner to amateur athletes. Therefore, training frequencies and patterns reported here may be reflective of the majority of MMA athletes. Our data only includes subjective internal measures without external load being quantified. Further studies should use reliable technology [61] to determine the external load of MMA training and its relationship to subjective internal measures. Studies should also be conducted to determine effects of different tapering strategies on both performance and weight making prior to competition. It is of key importance to note, however, that physiological requirements of and responses to MMA competition are largely unquantified [35]. This leaves a void in the planning of training due to a lack of known physiological targets to be achieved. Investigations of the direct internal and external load of MMA competition should be therefore prioritised in order to make optimal use of the data presented here.

In conclusion, we report for the first time typical weekly and daily training load and periodisation strategies employed by MMA athletes. Such information has not previously been provided in the literature, an omission which may prevent effective training and performance models from being developed or utilised [35]. In contrast to our hypothesis, we observed limited evidence of training periodisation within or between weekly microcycles. Additionally, differences in training load practices between athletes preparing for competition and athletes engaged in 'normal' training was limited to the final week before competition and was abrupt and stepwise, largely due to reduced training duration. We also observed no changes in ratings of fatigue or soreness throughout the 8-week observational period, or prior to bouts. From a practical perspective, we provide intensity classifications for MMA training categories which may be used to manipulate weekly load and pre-competition tapers. Using these data, researchers may now work with coaches to determine the optimal loading and tapering strategies to ensure appropriate physiological fatigue-recovery-adaptation alongside skill learning for competitive MMA athletes.

## Practical applications

We present relative intensities of MMA training modes for the first time and demonstrate that current MMA training practices do not follow a current understanding of optimal athlete competition preparation. These data may be used by MMA coaches to better plan and distribute training, providing weeks of progressive overreaching followed by weeks of restitution through their technical training sessions, in keeping with periodisation theory. This study also demonstrates the null effect of a one-week step taper on fatigue amongst MMA athletes, which should instigate further studies into the effects of different tapering strategies on this population. Finally, these data should be used by coaches to more appropriately distribute training intensity to promote the physiological responses of training at low-moderate intensities, whilst mitigating the negative effects of extended high intensity training.

## Acknowledgments

The authors sincerely thank all the MMA competitors, coaches and gyms for providing their resources, time, effort and support to this project.

## Author Contributions

**Conceptualization:** Christopher Kirk, Carl Langan-Evans, David R. Clark, James P. Morton.

**Data curation:** Christopher Kirk.

**Formal analysis:** Christopher Kirk.

**Investigation:** Christopher Kirk, Carl Langan-Evans, David R. Clark, James P. Morton.

**Methodology:** Christopher Kirk, Carl Langan-Evans, David R. Clark, James P. Morton.

**Project administration:** Christopher Kirk.

**Software:** Christopher Kirk.

**Supervision:** Carl Langan-Evans, David R. Clark, James P. Morton.

**Writing – original draft:** Christopher Kirk.

**Writing – review & editing:** Christopher Kirk, Carl Langan-Evans, David R. Clark, James P. Morton.

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
