## [Decision Letter · Decision Letter 0]

22 Mar 2021

PONE-D-21-00361

Quantification of training load distribution in mixed martial arts athletes: A lack of periodisation and load management

PLOS ONE

Dear Dr. Kirk,

Thank you for submitting your manuscript to PLOS ONE. After careful consideration, we feel that it has merit but does not fully meet PLOS ONE’s publication criteria as it currently stands. Therefore, we invite you to submit a revised version of the manuscript that addresses the points raised during the review process.

We look forward to receiving your revised manuscript.

Kind regards,

Cristina Cortis, Ph.D.

Academic Editor

PLOS ONE

Journal Requirements:

2. Thank you for including your ethics statement:  "Ethical approval was provided by the research ethics committee of Liverpool John Moores University (Ref: 19/SPS/007)".  

a. Please provide additional details regarding participant consent. In the ethics statement in the Methods and online submission information, please ensure that you have specified (i) whether consent was informed and (ii) what type you obtained (for instance, written or verbal, and if verbal, how it was documented and witnessed). If your study included minors, state whether you obtained consent from parents or guardians. If the need for consent was waived by the ethics committee, please include this information.

Reviewers' comments:

Reviewer's Responses to Questions

**Comments to the Author**

1. Is the manuscript technically sound, and do the data support the conclusions?

Reviewer #1: Yes

Reviewer #2: Partly

2. Has the statistical analysis been performed appropriately and rigorously? 

Reviewer #1: Yes

Reviewer #2: Yes

3. Have the authors made all data underlying the findings in their manuscript fully available?

Reviewer #1: Yes

Reviewer #2: Yes

4. Is the manuscript presented in an intelligible fashion and written in standard English?

Reviewer #1: Yes

Reviewer #2: Yes

5. Review Comments to the Author

Reviewer #1: This article represents a valuable work for the entire research on monitoring load in martial arts. Nevertheless, some comments are needed to achieve an improvement.

Abstract

Line 32: What’s the parameter/measurement used express the “weekly mean range” of load? It should be reported (Session-RPE or others?). The same also for the other parameters (in particular, no result has been reported for monotony and strain).

Lines 42-44. This conclusion is too strong. These data represent only “an” MMA experience but not “THE” MMA experience.

Introduction

No references and explanations have been provided in relation to the measurements adopted in the study. In particular, some words should be reported for session-RPE, short questionnaire of fatigue, and body region soreness by means of a CR10 scale if they have been already used in previous study in MMA, other martial arts, or in general.

For example, session-RPE has been already used to quantify internal training load in two different youth taekwondo training on the same athletes (Lupo et al., 2017): one focused on fundamental techniques; one focused on competitive/fighting workouts (reporting different outcomes).

Lupo C., Capranica L., Cortis C., Guidotti F., Bianco A., Tessitore A. (2017). Session-RPE for quantifying load of different youth taekwondo training sessions. Journal of Sports Medicine and Physical Fitness. 57(3), 189-194.

Method

Experimental design

Lines 104-105. Even if the fact that no intervention has been provided on training by the authors, it could be again reported in this period. For example: “All MMA training sessions were attended in person by the lead author (with no intervention/only to collect experimental data)….”

Participants

Even though participants to the study took part in this study following institutional ethical approval, it could be supposed that they signed an informed consent form too. This fact should be reported in the manuscript.

Results are well reported with clear subsections.

Discussion section is rationally reported, highlighting the main findings first, and specific interpretations after.

Practical applications have been exhaustively reported.

Reviewer #2: Kirk and co-workers submitted a manuscript entitled “Quantification of training load distribution in mixed martial arts athletes: A lack of periodization and load management”. They evaluated the training load and periodization strategies in MMA athletes, an interesting popultion given the scarcity of data available in the literature. The study may have the potential to add meaningful information to the current body of literature and is well written. I have some revisions to suggest in order to improve the manuscript quality:

Abstract: Descriptive statistics are missing. In addition, the article would benefit from the addition of statistical details related to the study’s results.

Introduction:

How the answer to this question is important to the field as this is not clear or obvious? How is this study and impactful study and not trivial as this needs more clarity as well. The key issue here is to make sure you set up your approach to the problem. In this form, the authors do not convince me of the real need for this study.

Methods

Line 119: I suggest adding descriptive statistics regarding the selected sample.

For each instrument used it is necessary to specify the brand, manufacturing company, and country.

Table 1. All acronyms should be followed by full names in the notes. What does Habitual Mass mean?

Figures overall: In several figures it is not specified what the bars represent. Standard error or 95% CI? I would be interested in seeing the 95CI for within-subject comparisons.

My main concern is the sample size selection. For scientific reasons, the sample size for trial needs to be planned carefully. The authors did not perform a power analysis to determine the sample size for the study, which appears to be quite small and not very representative of the populations chosen; also, the age differs substantially among the groups and sex interaction was not assessed or reported. In your case, if you want to detect a large effect size in your primary, considering the following design specifications: α= 0.05; (1-β) = 0.8; effect size f = 0.25, the sample size should be at least of 25 subjects. At this point, ES should be used to conduct a posteriori power analysis to justify your sample size.

The discussion section is very descriptive and offers limited comparisons to previous research.

Several key studies from the past 10 years regarding training load are missing. Similarly, how do practitioner benefit from that? Again, the discussion section fails to relate the findings to this particular application of interest. Authors are therefore encouraged to make substantial changes throughout introduction and discussion sections. In the current form the rationale for the study is not clear and the new value is unclear.

Overall, the study may have the potential to add meaningful information to the current body of literature.

6. PLOS authors have the option to publish the peer review history of their article (what does this mean?). If published, this will include your full peer review and any attached files.

Reviewer #1: **Yes: **Corrado Lupo

Reviewer #2: No

---

## [Author Response · Author response to Decision Letter 0]

9 Apr 2021

PONE-D-21-00361

Quantification of training load distribution in mixed martial arts athletes: A lack of periodisation and load management

PLOS ONE

Dear Dr. Kirk,

Thank you for submitting your manuscript to PLOS ONE. After careful consideration, we feel that it has merit but does not fully meet PLOS ONE’s publication criteria as it currently stands. Therefore, we invite you to submit a revised version of the manuscript that addresses the points raised during the review process.

Thank you for this suggestion. We completed this process, but this system resulted in our figures being converted into lower quality formats leading to some being cut in half and others having details obscured. We understand the need for some quality control of figures, but the figures provided have been prepared in Prism as field standard and aligned on the manuscript in the specific layout as intended, so we do not wish to alter this with third party graphical autorecognition software. 

We look forward to receiving your revised manuscript.

Kind regards,

Cristina Cortis, Ph.D.

Academic Editor

PLOS ONE

Journal Requirements:

Thank you for signposting us to this information, we have now made these formatting changes.

2. Thank you for including your ethics statement: "Ethical approval was provided by the research ethics committee of Liverpool John Moores University (Ref: 19/SPS/007)". 

a. Please provide additional details regarding participant consent. In the ethics statement in the Methods and online submission information, please ensure that you have specified (i) whether consent was informed and (ii) what type you obtained (for instance, written or verbal, and if verbal, how it was documented and witnessed). If your study included minors, state whether you obtained consent from parents or guardians. If the need for consent was waived by the ethics committee, please include this information.

Thank you for pointing out this omission. The manuscript has now been amended to include the statement: 

“A cohort of 14 competitive MMA participants from 4 individual MMA clubs volunteered to take part in this study following written, informed consent and institutional ethical approval based on the following inclusion criteria:…”

The statement on the submission form has also been amended to:

“Ethical approval was provided by the research ethics committee of Liverpool John Moores University (Ref: 19/SPS/007), with participants providing written, informed consent prior to commencement of data collection". 

Thank you for this clarification, we do not wish to change our statement at this time.

5. Review Comments to the Author

Reviewer #1: This article represents a valuable work for the entire research on monitoring load in martial arts. Nevertheless, some comments are needed to achieve an improvement.

To Dr Lupo,

Thank you for your measured and very helpful review of our work. We feel these changes have improved the quality of our manuscript and we hope that these changes meet your standards for publication.

Kind regards,

Christopher Kirk

Abstract

Line 32: What’s the parameter/measurement used express the “weekly mean range” of load? It should be reported (Session-RPE or others?). The same also for the other parameters (in particular, no result has been reported for monotony and strain).

Thank you for highlighting. We have now added these points to the abstract, and this section now reads:

“Using Bayesian analyses (BF10≥3), data demonstrate that training duration (weekly mean range = 3.9-5.3 hours), sRPE (weekly mean range = 1,287–1,791 AU), strain (1,143–1,819 AU), monotony (0.63-0.83 AU), fatigue (weekly mean range = 16-20 AU) and soreness did not change within or between weeks.”

Lines 42-44. This conclusion is too strong. These data represent only “an” MMA experience but not “THE” MMA experience.

We agree that we need to avoid over-interpreting our data and applying sample study results onto the entire population, so we have amended this statement to read:

“We conclude that periodisation of training load was largely absent in these periods of MMA training, as is the case within and between weekly microcycles.” 

Introduction

No references and explanations have been provided in relation to the measurements adopted in the study. In particular, some words should be reported for session-RPE, short questionnaire of fatigue, and body region soreness by means of a CR10 scale if they have been already used in previous study in MMA, other martial arts, or in general.

For example, session-RPE has been already used to quantify internal training load in two different youth taekwondo training on the same athletes (Lupo et al., 2017): one focused on fundamental techniques; one focused on competitive/fighting workouts (reporting different outcomes).

Lupo C., Capranica L., Cortis C., Guidotti F., Bianco A., Tessitore A. (2017). Session-RPE for quantifying load of different youth taekwondo training sessions. Journal of Sports Medicine and Physical Fitness. 57(3), 189-194.

Thank you for highlighting this discrepancy in our introduction. We have now added a penultimate paragraph to the introduction and some smaller statements throughout the introduction to provide information about these data collection methods with examples of their previous, supported use in applied sports research.

Method

Experimental design

Lines 104-105. Even if the fact that no intervention has been provided on training by the authors, it could be again reported in this period. For example: “All MMA training sessions were attended in person by the lead author (with no intervention/only to collect experimental data)….”

We have now altered this statement to read:

“All MMA training sessions were attended in person by the lead author for the purposes of data collection without intervention to the training sessions themselves.”

Participants

Even though participants to the study took part in this study following institutional ethical approval, it could be supposed that they signed an informed consent form too. This fact should be reported in the manuscript.

Thank you for highlighting this oversight on our part. This statement has now been amended to read:

“A cohort of 14 competitive MMA participants from 4 individual MMA clubs volunteered to take part in this study following written, informed consent and institutional ethical approval based on the following inclusion criteria….”

Results are well reported with clear subsections.

Discussion section is rationally reported, highlighting the main findings first, and specific interpretations after.

Practical applications have been exhaustively reported.

Thank you for these responses, we are very pleased to see that the presentation of results and our interpretations are logical, easy to follow and lead the reader to our conclusions.

Reviewer #2: Kirk and co-workers submitted a manuscript entitled “Quantification of training load distribution in mixed martial arts athletes: A lack of periodization and load management”. They evaluated the training load and periodization strategies in MMA athletes, an interesting popultion given the scarcity of data available in the literature. The study may have the potential to add meaningful information to the current body of literature and is well written. I have some revisions to suggest in order to improve the manuscript quality:

To the Reviewer,

Thank you for taking the time to review our manuscript and provide the comments below. We believe that these comments have helped improve our work and we hope that the changes made are sufficient to warrant publication in PLOS One.

Kind regard,

Christopher Kirk

Abstract: Descriptive statistics are missing. In addition, the article would benefit from the addition of statistical details related to the study’s results.

Thank you for highlighting the omission of descriptive statistics regarding our cohort, these have now been added into the abstract. In terms of statistical detail, our study consists of a large number of statistical analyses. Reporting each of these in the abstract would take up most of the permitted word count and would not allow us to describe the study itself. Therefore, we have opted to include further weekly mean ranges of our variables and we draw the reviewer’s attention to the inclusion of the Bayes factor acceptance standard (BF10≥3) in the original manuscript abstract to indicate the minimal strength of the results.

Introduction:

How the answer to this question is important to the field as this is not clear or obvious? How is this study and impactful study and not trivial as this needs more clarity as well. The key issue here is to make sure you set up your approach to the problem. In this form, the authors do not convince me of the real need for this study.

We have now included further statements in the introduction and discussion to make the point clear that no training load data for the sport of MMA currently exists, and the consequences of this for coaches, athletes and researchers. We have also included additional statements in the discussion to demonstrate that this study is unique and novel in that it reports the training load and fatigue responses of a sport for the first ever time and how these data may be used by multiple stakeholders, further supporting the impact and importance of the study as a whole. 

Methods

Line 119: I suggest adding descriptive statistics regarding the selected sample.

Thank you for bringing this potential source of confusion to our attention. Participant descriptive data are provided in Table 1, which we have now made clear for the reader in the opening sentence of this paragraph.

For each instrument used it is necessary to specify the brand, manufacturing company, and country.

The authors apologise, as we are unsure which specific tools the reviewer is referring to, as all the data collection tools were paper based questionnaires, which have been referenced throughout. The only physical tool used in this study was the video camcorder, the details of which were listed on line 163 of the original manuscript and are now found on line 187 of the revised manuscript.

Table 1. All acronyms should be followed by full names in the notes. What does Habitual Mass mean?

Thank you for highlighting this potential source of confusion for the reader. Habitual mass is a common term in weight-making sports meaning the participant’s body mass before commencing body mass reduction, but we accept that this term might not be known to readers from non-weight making sports. To amend this we have changed the column heading to read “Habitual Body Mass” and defined this in the table footer to make this clear.

Figures overall: In several figures it is not specified what the bars represent. Standard error or 95% CI? I would be interested in seeing the 95CI for within-subject comparisons.

Thank you for highlighting this oversight on our part. Labels have now been added to show that error bars in each figure show 95% credible intervals or standard deviations where appropriate. Within participant comparisons have not been completed in this study as it is outside the aims or scope of the study as a whole. 

My main concern is the sample size selection. For scientific reasons, the sample size for trial needs to be planned carefully. The authors did not perform a power analysis to determine the sample size for the study, which appears to be quite small and not very representative of the populations chosen; also, the age differs substantially among the groups and sex interaction was not assessed or reported. In your case, if you want to detect a large effect size in your primary, considering the following design specifications: α= 0.05; (1-β) = 0.8; effect size f = 0.25, the sample size should be at least of 25 subjects. At this point, ES should be used to conduct a posteriori power analysis to justify your sample size.

The authors appreciate the potential confusion regarding power in relation to Bayes factors. The analyses conducted and presented are Bayesian analyses, meaning any type of frequentist power analyses (a priori or post hoc) based on effect sizes would be unrelatable to the analyses conducted and therefore largely meaningless (Kruschke & Liddell, 2018; Wagenmakers et al., 2015). Frequentist power analyses are applied to determine the likelihood of finding the specified effect size in hypothetical repeated trials of the same experiment. Bayesian analyses compare the strength of H1 against H0 using the existing data regardless of sample size or hypothetical repeat trials, meaning no hypothetical data are required (Morey, Romeijn, & Rouder, 2014). The strength of Bayesian analyses are instead based on the prior, which in this case was an informed JZS prior r = .707, as recommended for such analyses where no previous data exist, as is the case here (Kruschke & Liddell, 2018; Wagenmakers et al., 2015). In keeping with recommendations around the precision of default informed priors, and as mentioned in the methods, robustness checks were performed on each test with each reported where required in the results (van de Schoot & Depaoli, 2014; van Doorn et al., 2019; Wagenmakers et al., 2018). 

Use of a power analysis based on an expected effect size to determine the required sample size would also be inappropriate as this was not an experimental trial, so no expected effect size was specified. In order to specify an effect size a priori, there would need to be extant data from previous studies using the variables and the conditions in the study. As stated in the introduction, no such data exists in MMA, so there is no reason to believe that an effect size of 0.25 would be expected or required. In this instance effect sizes of 0.1, 0.25, 0.4 and 0.8 may all be equally expected and likely. For this reason, this study was specifically designed as an observational study to measure the training load changes of MMA for the first time, with no prior effect size expectations. Whilst sample sizes for experimental trials using Bayesian analyses can be chosen a priori using BFDN (Stefan, Gronau, Schӧnbrodt, & Wagenmakers, 2019), this study is not an experimental trial, meaning this process would not be appropriate either. 

The authors believe that the sample included is representative of both male and female MMA competitors based on the ages, mass and stature reported in previous studies performed on this population (Alm & Yu, 2013; Andrade, Junior, Andreato, & Coimbra, 2018; Bodden, Needham, & Chockalingam, 2015; Hurst, Atkins, & Kirk, 2014; Kirk, Hurst, & Atkins, 2015; Marinho, Del Vecchio, & Franchini, 2011; MG Schick et al., 2010; Monica Schick, Brown, & Schick, 2012). From a practical application standpoint, we argue that the sample is representative as it is made up of the available competitive MMA participants who fit the inclusion criteria listed, all of whom were either professional or world level amateurs at the time of data collection. As the aim of the study is to understand the training practices of MMA competitors, this sample is directly representative of such a group. 

The discussion section is very descriptive and offers limited comparisons to previous research.

Several key studies from the past 10 years regarding training load are missing. Similarly, how do practitioner benefit from that? Again, the discussion section fails to relate the findings to this particular application of interest. Authors are therefore encouraged to make substantial changes throughout introduction and discussion sections. In the current form the rationale for the study is not clear and the new value is unclear.

As an observational study, the discussion will always have a descriptive element. We have endeavoured, however, to relate our data and findings to theoretical explanations and comparisons to other studies from combat sports and other sports throughout. The introduction and discussion demonstrate that there is extremely limited research into MMA training load to compare against, but what does exist we feel we have discussed in detail. To bridge this gap in sport specific research we have supported our discussions and overall conclusions with a wide range of training load, physiology and performance research spanning not only the last 10 years, but also the last 20 years (evidenced by the large number of references included in the manuscript as a whole). If the reviewer feels there are essential papers related to the topic of training load measurement or distribution that must be included, then we respectfully request that they provide these studies with reference to the specific sections of our discussion that they relate to.

We have, however, altered some of our statements at the start and end of the discussion to make it clearer for the reader about the aims and nature of the study as whole. We did not wish to alter large portions of our discussion as Reviewer 1 stated that this part of our manuscript was detailed and strong. Therefore, changing too much of this section would potentially change their recommendations for our paper, especially in light of Reviewer 2 not providing any specific requests to amend any specific sections or details of the discussion.

Overall, the study may have the potential to add meaningful information to the current body of literature.

Thank you for this comment.

---

## [Decision Letter · Decision Letter 1]

23 Apr 2021

Quantification of training load distribution in mixed martial arts athletes: A lack of periodisation and load management

PONE-D-21-00361R1

Dear Dr. Kirk,

We’re pleased to inform you that your manuscript has been judged scientifically suitable for publication and will be formally accepted for publication once it meets all outstanding technical requirements.

Kind regards,

Cristina Cortis, Ph.D.

Academic Editor

PLOS ONE

Reviewers' comments:

Reviewer's Responses to Questions

**Comments to the Author**

1. If the authors have adequately addressed your comments raised in a previous round of review and you feel that this manuscript is now acceptable for publication, you may indicate that here to bypass the “Comments to the Author” section, enter your conflict of interest statement in the “Confidential to Editor” section, and submit your "Accept" recommendation.

Reviewer #1: (No Response)

Reviewer #2: All comments have been addressed

2. Is the manuscript technically sound, and do the data support the conclusions?

Reviewer #1: (No Response)

Reviewer #2: Yes

3. Has the statistical analysis been performed appropriately and rigorously? 

Reviewer #1: (No Response)

Reviewer #2: Yes

4. Have the authors made all data underlying the findings in their manuscript fully available?

Reviewer #1: (No Response)

Reviewer #2: Yes

5. Is the manuscript presented in an intelligible fashion and written in standard English?

Reviewer #1: (No Response)

Reviewer #2: Yes

6. Review Comments to the Author

Reviewer #1: (No Response)

Reviewer #2: (No Response)

7. PLOS authors have the option to publish the peer review history of their article (what does this mean?). If published, this will include your full peer review and any attached files.

Reviewer #1: **Yes: **Corrado Lupo

Reviewer #2: No

---

## [Editor Report · Acceptance letter]

29 Apr 2021

PONE-D-21-00361R1 

Quantification of training load distribution in mixed martial arts athletes: A lack of periodisation and load management 

Dear Dr. Kirk:

I'm pleased to inform you that your manuscript has been deemed suitable for publication in PLOS ONE. Congratulations! Your manuscript is now with our production department. 

Kind regards, 

on behalf of

Prof. Dr. Cristina Cortis 

Academic Editor

PLOS ONE